# Impact of Hierarchy on Multidisciplinary Heart-Team Recommendations in Patients with Isolated Multivessel Coronary Artery Disease

**DOI:** 10.3390/jcm8091490

**Published:** 2019-09-19

**Authors:** Mohamed Abdulrahman, Alaa Alsabbagh, Thomas Kuntze, Bernward Lauer, Marc A. Ohlow

**Affiliations:** 1Division of Cardiology, Zentralklinik, Bad Berka 99437, Germany; 2Division of Cardiovascular Surgery, Zentralklinik, Bad Berka 99437, Germany; 3Division of Cardiology, University Jena, Jena 07742, Germany

**Keywords:** hierarchy, heart team, multivessel disease, coronary, intervention

## Abstract

**Background:** The Heart Team (HT) discussion has been incorporated in the current guidelines for myocardial revascularization in order to optimize treatment decisions for patients with multivessel coronary disease (MVD). There are no data in the literature, whether hierarchical issues do have an impact on HT decisions. We aimed to analyze the therapeutic recommendations of the multidisciplinary “Heart Team” (HT) for coronary artery bypass grafting (CABG) or percutaneous coronary intervention (PCI) if: (a) The head of cardiovascular surgery (HOS) and the head of cardiology (HOC) were present during the HT meeting, (b) both directors were absent, (c) only HOS or HOC was present. **Methods:** Retrospective analysis of all HT discussions between 2012 and 2015 in patients with isolated MVD (without any other cardiac problems requiring surgery). **Results:** During the study period, we analyzed 209 HT discussions in patients with isolated MVD. If neither HOS nor HOC was present at the HT discussion, the therapeutic recommendation was in 69% CABG and 31% PCI. If HOS and HOC were present in 77% CABG and 23% PCI was recommended (*p* = 0.34). If only HOS was present therapeutic recommendation was in 83% CABG and 17% PCI, and if only HOC was present the recommendation was in 54% CABG and 46% PCI (*p* < 0.0001). This difference did not attenuate during the study period. **Conclusions:** The hierarchy of the participating physicians significantly impacts treatment recommendations of a multidisciplinary HT in patients with isolated MVD. This impact did not attenuate after several years of Heart Team interaction.

## 1. Introduction

A team concept in medicine was first established in oncology and organ transplant programs [1]. Tumor boards have been making multi-specialty disease management decisions in oncology since the 1950s [2]. Initiated in early randomized trials comparing coronary artery bypass grafting (CABG) with medical therapy for stable coronary artery disease (CAD), a Heart Team was used to select patients eligible for randomization [3]. Partly due to the introduction of percutaneous coronary intervention (PCI), interventional cardiologists, and cardiac surgeons were increasingly targeting the same patient population [3]. The specific term “Heart Team” (HT) describes specialists working together to optimize treatment recommendations based on the exchange of knowledge and experience with specific therapies [3]. The Heart Team concept has lately gained increasing attraction in the context of broad range procedures like PCI, transcatheter valve replacement/repair, and other complex cardiovascular interventions [1]. The concept has also been incorporated in the European and American guidelines on myocardial revascularization as a class I C recommendation [4,5].

Several studies and publications have analyzed the decision-making process of the HT, the outcome, and the reproducibility of the decisions [6,7]. Other authors have analyzed the patients’ preferences for CABG or staged PCI in multivessel CAD [8]. However, there are overt and subconscious factors that influence the decision for a revascularization strategy [3], among them conflict of interest with industry, not being up-to-date with CABG/PCI, personal conflict between cardiologist and surgeon, and “turf protection” [3].

The primary objective of our study was to analyze the therapeutic recommendations of the multidisciplinary HT (CABG versus PCI) in patients with isolated multivessel disease. We analyzed the impact of the following scenarios: (a) The director of the department of cardiovascular surgery and the director of the department of cardiology were present during the HT meeting, (b) both department directors were absent, (c) only the director of one department (cardiovascular surgery or interventional cardiology) was present.

## 2. Methods

HT meetings are convened each business weekday at 3 p.m. in our tertiary referral coronary intervention/cardiac surgery unit. At a minimum, a board-certified cardiac surgeon, an interventional cardiologist, and one non-interventional cardiologist were present at each meeting. No other medical disciplinaries are usually involved in the discussion, and also patients or their relatives are not present during the HT meeting. The meeting is chaired by a consultant (cardiac surgeon or cardiologist), and the HT coordinator documents the decisions. Meetings are also attended by junior medical and surgical staff and sixth-year students.

In a given patient, a diagnostic coronary angiography is performed and, if (1) a single or two-vessel disease is diagnosed, the therapeutic decision is left to the discretion of the cardiologist in the Cath lab and is in most cases, stenting in the same procedure. If (2) a two-vessel disease with significant left main stenosis or a three-vessel disease is diagnosed, the patient is taken off the table and the case is discussed in the next HT meeting (usually on the same day in the afternoon). There is an HT proforma which is filled-in before the meeting so that the relevant data are at hand during the meeting. Prior to the meeting the resident responsible for the patient ensures that appropriate information (patient details, comorbidities are completed on the HT proforma, coronary angiography films, echocardiography recordings, results of stress tests, SYNTAX (SYNergy between percutaneous coronary intervention with TAXus and cardiac surgery [9]) and EuroScore II calculations are also available for the meeting. The decisions are noted on the individual patient’s proforma sheet and documented in the patient’s notes.

All patients at our center between February 2012 and December 2015 undergoing left-sided heart catheterization and met the following criteria were included in our study: (1) Unprotected left main CAD, (2) three-vessel disease, or (3) two-vessel disease including a significant lesion of the main stem or the proximal left anterior descending artery. HT discussions on patients with concomitant valvular disease or other cardiac problems requiring surgery were excluded. Moreover, patients with acute coronary syndrome and multivessel coronary artery disease needing emergency treatment were not discussed in the HT. The retrospectively gathered data were analyzed. No informed consent was required. This study was approved by the institutional review board of the Zentralklinik Bad Berka.

The presence of the department directors at the HT meetings was determined by their presence in the hospital and their busy schedules. Therapeutic recommendations of patients with very complex coronary anatomy were always given at the meeting when their cases were presented. None of such complex patients were presented exclusively to the department directors so that there is no “presentation bias” affecting our results. After a treatment decision is reached, the patients are informed, patient’s preference is taken into account. Patient consent is obtained and, when applicable, the patient is scheduled for the procedure.

### Statistical Analysis

Simple descriptive statistics were used to describe the study population. The difference between the Heart Team decisions was compared using the χ^2^- or Fisher-exact test. One-way analysis of variance (ANOVA) was used to compare continuous variables of unmatched groups. A probability value of <0.05 was considered to be statistically significant. The appropriateness of the HT decisions for CABG or PCI according to AUC of the American College of Cardiology and according to the SYNTAX score II depending on the presence or absence of the department directors is shown in Section 3. Statistical analysis was performed using GraphPad Prism version 6.02 for Windows (GraphPad Software, La Jolla, CA, USA).

## 3. Results

During the study period, a total of 306 HT-discussions could be analyzed, among them 209 (68.3%) discussing patients with isolated multivessel disease. Among those 209 discussed patients, there were 174 (83.3%) males, 18 (8.6%) had undergone prior cardiac surgery, and the prevalence of cardiovascular risk factors was high. Patients undergoing PCI had more frequently a history of cardiac surgery and chronic obstructive pulmonary disease, had a higher operative risk as determined by a higher EuroScore II, and a lower left ventricular ejection fraction. Patients undergoing CABG had more complex coronary anatomy with significantly higher SYNTAX-Scores. The baseline characteristics of the patients according to different HT composition are presented in Table 1.

Decisions taken included medical management in nine (4.3%), PCI in 59 (28.1%), and CABG in 141 patients (67.6%). During 126 HT meetings (60.0%) at least one of the department directors was present. When both of the department directors were absent, the recommendation of the HT was in 69% CABG and 31% PCI (CABG-to-PCI ratio 2.23). When both department directors were present, in 77% of the cases cardiac bypass surgery was recommended and in 23% PCI resulting in a CABG-to-PCI ratio of 3.35 (the *p*-value for HT decisions in the absence versus in the presence of both department directors was 0.34). If only the director of the department of cardiovascular surgery attended the HT meeting, 83% of the patients were scheduled for CABG, and only 17% for PCI (CABG-to-PCI ratio 4.88). Finally, when only the director of the department of cardiology was present, 54% of the discussed cases were listed for cardiac bypass surgery and 46% for PCI resulting in a CABG-to-PCI ratio of 1.17 (*p*-value < 0.0001 for HT decisions when only one of the department directors was present), as shown in Figure 1 and Figure 2.

In order to assess the development of the HT decision making over time, we compared the results of the early phase after establishing the HT concept in our hospital with the results after several years of HT interaction (years 2012–2015). However, we were not able to demonstrate attenuation of the marked differences between HT decisions being made in the presence of only the director of the cardiovascular surgery or the director of interventional cardiology (Figure 3).

### Application of the American College of Cardiology “Appropriate Use Criteria” and “SYNTAX Score II”

The final HT recommendation was adjudicated for appropriateness using the American College of Cardiology appropriate use criteria (AUC) [10] and calculation of the SYNTAX score II [10,11] for each case. Out of the 209 HT recommendations, 181 (86.6%) could be reviewed. Nine patients (4.3%) were treated medically and therefore excluded. In the remaining 19 patients (9.1%), their clinical data were incomplete and prevented a review of the HT decision. 

The appropriateness of the HT decisions for CABG or PCI according to AUC of the American College of Cardiology and according to the SYNTAX score II depending on the presence or absence of the department directors is shown in Table 2 and Figure 4.

## 4. Discussion

Our study, including patients with a three-vessel disease or two-vessel disease plus a significant lesion of the main stem, demonstrated that multidisciplinary HT decisions are not only influenced by the current guideline recommendations. HT decisions also seemed to be strongly influenced by hierarchy among the members of the HT.

There is a marked diversity of practice patterns between countries concerning the application of CABG and PCI [12]. For example, in patients with three-vessel disease, PCI was performed in 30% (CABG-to-PCI ratio: 2.3) in Europe, but only 17% (CABG-to-PCI ratio: 4.9) in North America [13]. Within Europe, rates of CABG for three-vessel disease ranged from 42% (CABG-to-PCI ratio: 0.7) in France to 90% (CABG-to-PCI ratio: 9) in the United Kingdom [13]. The Organization of Economic Collaboration and Development (OECD) reported a mean CABG-to-PCI ratio of 0.30 for 1, 2 or 3 vessel disease in 2015 in those countries affiliated with this organization, ranging from a low of 0.12 to a high of 1.49 [14]. Even in the same health-care system, a significant difference in CABG-to-PCI ratio was observed across different regions [14]. This wide variability in the type of revascularization utilization might be the result of differences in baseline characteristics [3] but also driven by economic and reimbursement considerations [15].

There is only one analysis available in the literature on variations in clinical decision-making between cardiologists and cardiac surgeons on a hospital-level [16] from 2006 before the introduction of the HT concept. The authors were able to demonstrate that there is a poor agreement between cardiac clinical specialists (cardiac surgeons, interventional cardiologists, and non-interventional cardiologists) in the choice of treatment offered to the patients [16], and this variation was most significant between surgeons and interventional cardiologists [16]. These results supported findings of other areas of clinical medicine where clinical specialists who perform a procedure are more likely to consider it appropriate in specific case scenarios [17]. Studies have shown that in 68% of patients undergoing PCI and 59% of those who underwent CABG, the alternative revascularization strategy was not discussed with the patient [18]. A multidisciplinary team and HT approaches were established to increase agreement among surgeons and cardiologists concerning the choice of treatment [3,16]. However, our study showed evidence that even after establishing the multidisciplinary HT concept in our tertiary referral hospital, the choice of treatment is influenced by the composition and the “clinical pecking order” of the HT. When only the director of the department of cardiovascular surgery was present at the HT meeting, the CABG-to-PCI ratio in multivessel disease was on average 4.88 versus 1.17 when only the director of the department of cardiology was present (*p* < 0.0001). This difference did not attenuate even after several years of HT interaction, which is in some contrast to evidence suggesting that the longer an HT has worked together, the more interactive, and successful it becomes [3]. However, if none or if both of the directors were present at the HT meeting, the CABG-to-PCI ratio was on average 2.70 without statistically significant differences over the years.

The HT concept is in general well accepted in Germany, and further proof for this is that recently a cardiac surgeon served as the president of the German Cardiology Annual Meeting and a cardiologist as the President of the German Cardiac Surgery Annual Meeting. However, one author referred to the HT concept as a “Platonic illusion”, a kind of fashionable euphemism [1].

### 4.1. Implications for Daily Practice

A well-known fact from psychological studies of multidisciplinary teamwork is that, if healthcare professionals are taken out of their typical work environments, the impact of hierarchy and stereotypical behavior with “us” and “them” attitudes mostly dissolve [19,20]. Possibly, HT discussions outside the context of a hospital might help to overcome issues related to hierarchical and stereotyped behaviors [20]. Second, ideally the primary carer for a patient, such as a community practitioner, should be involved in the HT discussion [16], as the debate for any multivessel CAD is frequently limited to CABG and PCI, oblivious to the fact that medical therapy has also advanced and does not shorten life expectancy in most of the cases [1]. However, in these possible solutions for issues with HT interaction, logistics remain a significant limitation, as members may not be available, all at the pre-arranged location and time. Thirdly, simulation-based training programs offer possibilities to simulate HT discussions and learn to apply guideline suggested strategies [10]. Finally, the SYNTAX score II for the prediction of mortality after CABG and PCI developed and validated in 2013 [11] adds clinical characteristics to the anatomical SYNTAX score to improve individualized treatment decisions [21]. Wider implementation of tools like the American College of Cardiology appropriate use criteria [5] or the SYNTAX score II [10,11] may further reduce the impact of hierarchy on HT decisions.

In a recent publication of the SYNTAX Extended Study (SYNTAXES), at ten years, no significant difference existed in all-cause death between PCI using first-generation paclitaxel-eluting stents and CABG [22]. However, CABG provided a significant survival benefit in patients with three-vessel disease, but not in patients with left main coronary artery disease. These results might change the HT decision making in future.

### 4.2. Limitations

This study should be interpreted given the following limitations. Firstly, limitations inherent to a retrospective observational single academic medical center study cannot be excluded. Secondly, the director of the department of cardiology and the director of the department of cardiovascular surgery at our institution have been personal friends for more than two decades. Therefore, the results obtained in our Heart Team might do not apply to other multidisciplinary teams dealing with patients with multivessel coronary disease. Finally, leadership cultures among different countries regarding attitudes toward authority can vary from egalitarian (e.g., Australia, United States of America) to hierarchical (e.g., China, Japan, Germany) [23] which may affect the transferability of our results to other countries. 

## 5. Conclusions

Our study shows that treatment recommendations of a multidisciplinary Heart Team in patients with isolated multivessel coronary artery disease are significantly impacted by the hierarchy of the participating physicians. This impact did not attenuate after several years of Heart Team interaction.

## Figures and Tables

**Figure 1 jcm-08-01490-f001:**
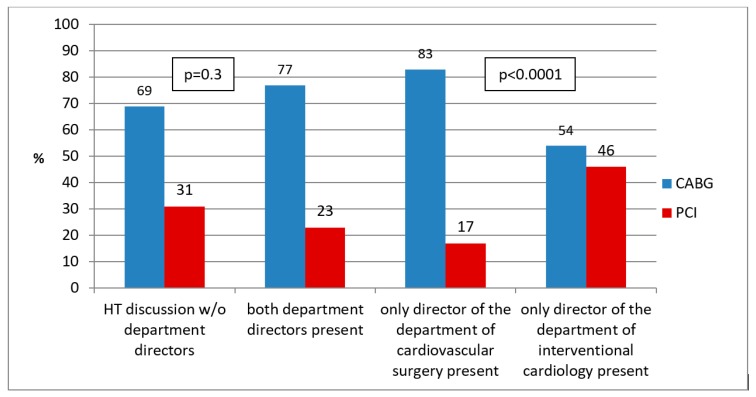
Heart Team decisions in relation to the absence or presence of the directors of the department of cardiovascular surgery or the department of interventional cardiology. Legend: Bar chart of patients either scheduled for CABG (**blue columns**) or PCI (**red columns**). HT = Heart Team; CABG = coronary artery bypass grafting; PCI = percutaneous coronary intervention; w/o = without.

**Figure 2 jcm-08-01490-f002:**
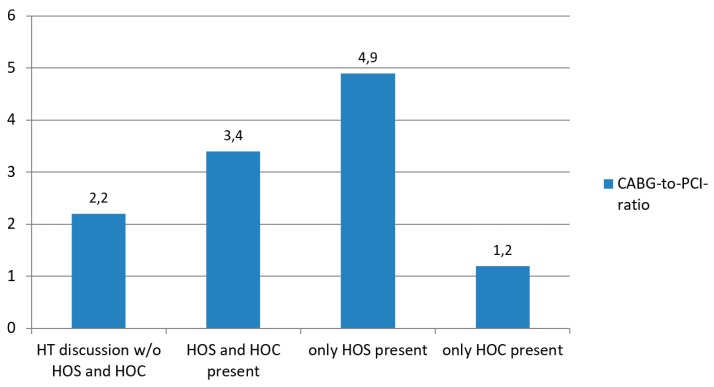
Heart Team decisions in relation to the absence or presence of the directors of the department of cardiovascular surgery or the department of interventional cardiology. Legend: Bar chart of CABG-to-PCI ratio. HOC = Head of Cardiology; HOS = Head of Cardiac Surgery; HT = Heart Team; CABG = coronary artery bypass grafting; PCI = percutaneous coronary intervention; w/o = without.

**Figure 3 jcm-08-01490-f003:**
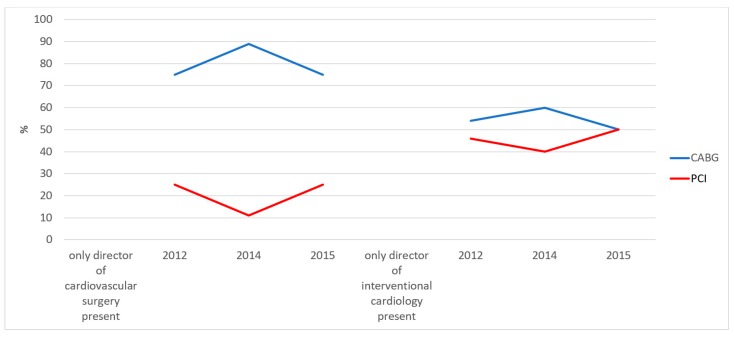
Evolution of Heart Team decisions over time related to the presence of the directors of the department of cardiovascular surgery (**left**) or the department of interventional cardiology (**right**). Legend: Bar chart of patients either scheduled for CABG (**blue line**) or PCI (**red line**). CABG = coronary artery bypass grafting; PCI = percutaneous coronary intervention.

**Figure 4 jcm-08-01490-f004:**
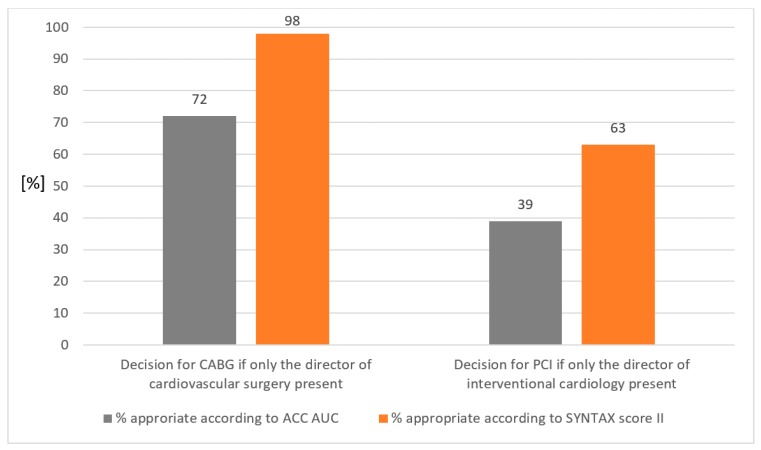
Appropriateness of Heart Team decisions. Legend: Bar chart of the appropriateness according to American College of Cardiology appropriate use criteria (ACC AUC: grey bar) and SYNTAX score II (yellow bar). CABG = coronary artery bypass grafting; PCI = percutaneous coronary intervention.

**Table 1 jcm-08-01490-t001:** Baseline characteristics.

Variable	HOS Present (*n* = 54)	HOC Present (*n* = 38)	HOC and HOS Present (*n* = 24)	Neither HOC nor HOS Present (*n* = 85)	*p*-Value
Age (years)	70.1 ± 9.6	69.3 ± 9.1	70.9 ± 10.1	70.5 ± 8.9	0.87
Male	43 (79.6)	34 (89.5)	19 (79.2)	69 (81.2)	0.61
**Comorbidities**					
Body-mass-index (kg/m^2^)	28 ± 8.3	28 ± 4.6	31.4 ± 7.1	29.6 ± 7.2	0.69
Hypertension requiring therapy	36 (66.7)	31 (81.6)	17 (70.8)	65 (79.3)	0.54
Active cigarette smoking	4 (7.4)	12 (31.6)	4 (16.7)	26 (30.6)	**0.006**
Hyperlipidemia	14 (25.9)	22 (57.9)	13 (54.2)	36 (42.4)	**0.01**
History of cardiac surgery	3 (5.6)	4 (10.5)	2 (8.3)	10 (11.8)	0.67
History of PCI	8 (14.8)	13 (34.2)	15 (60.0)	18 (21.2)	**<0.001**
PAD	18 (33.3)	10 (26.3)	3 (12.5)	10 (11.8)	**0.01**
Chronic renal dysfunction	8 (14.8)	5 (13.2)	4 (16.7)	10 (11.8)	0.92
COPD	1 (1.9)	4 (10.5)	1 (4.2)	7 (8.2)	0.77
History of stroke/TIA	4 (7.4)	2 (5.3)	1 (4.2)	3 (3.5)	0.49
LV-EF (%)	54 ± 9.9	51 ± 13	49.4 ± 14.8	48.9 ± 13.7	0.08

Values are mean ± SD or n (%), bold values denote significant values. Percentages might not sum to 100% as a result of rounding. COPD: chronic obstructive pulmonary disease; HOC: Head of Cardiology; HOS: Head of Cardiac Surgery; LV-EF: left ventricular ejection fraction; PAD: peripheral artery disease; PCI: percutaneous coronary intervention; TIA: transitory ischemic attack.

**Table 2 jcm-08-01490-t002:** Angiographic characteristics and appropriateness calculations.

**Variable**	**HOS Present (*n* = 54)**	**HOC Present (*n* = 38)**	**Neither HOC nor HOS Present (*n* = 85)**	***p*-Value**
Diabetes mellitus	20 (37.0)	17 (44.7)	42 (49.4)	0.56
eGFR (mL/min/1.73 m^2^)	47.3	44.9	48.2	0.89
3 vessels with lesions ≥ 50%	52 (96.3)	33 (86.8)	78 (91.7)	0.37
2 vessels with LM/proximal LAD	2 (3.7)	3 (7.9)	4 (4.7)	0.75
Unprotected LM	0 (0.0)	2 (5.3)	3 (3.5)	0.47
EuroSCORE II (points)	3.4 ± 2.0	4.6 ± 12	4.7 ± 7.9	0.23
SYNTAX score (points)	28 ± 8.3	27 ± 8.4	27.9 ± 10.1	0.81
**Decision for CABG/PCI Appropriate According to**				
ACC appropriate use criteria				
Rarely appropriate care	2/50 (4.0)	5/33 (15.2)	10/76 (13.2)	0.17
May be appropriate care	16/50 (32.0)	7/33 (21.2)	21/76 (27.6)	0.56
Appropriate care	32/50 (64.0)	21/33 (63.6)	45/76 (59.2)	0.83
SYNTAX score II calculation	44/50 (88.0)	32/33 (97.0)	63/76 (82.9)	0.09

Values are mean ± SD or *n* (%). Percentages might not sum to 100% as a result of rounding. ACC: American College of Cardiology; eGFR: estimated glomerular filtration rate; EuroSCORE: European System for Cardiac Operative Risk Evaluation; HOC: Head of Cardiology; HOS: Head of Cardiac Surgery; LAD: left anterior descending artery; LM: left main stem; SYNTAX: SYNergy between percutaneous coronary intervention with TAXus and cardiac surgery.

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
