# Peer review of "Impact of Hierarchy on Multidisciplinary Heart-Team Recommendations in Patients with Isolated Multivessel Coronary Artery Disease"

_jcm, 2019, doi:10.3390/jcm8091490_

Round 1
Reviewer 1 Report
This is an interesting retrospective posthoc registry analysis of decision making for PCI or CABG at heart team meetings in a single centre.
The authors conclude that the presence of the head of department for surgery or cardiology impacted on the decision making.
Whilst I applaud the essence of the point made here, there are very many confounding factors (some of which the authors acknowledge) with this study.
The analysis of this study was all performed retrospectively - it isnt presented clearly how many patients actually went through the process in the way stated i.e. taken off the table if anatomy was complex as described and discussed the same day; or if the discussion was deferred until the desired person was present at the heart team meeting. It isnt clear for how many patients complete data was not available or who were deemed too high risk for anything, or who needed same day treatment.2. Despite the statement of limitations, departmental hierarchy structures are inevitably complex with multiple relationships and conflicts between individual members. It is a valid and interesting point to make that the necessity for a heart team is not evidence-based and is inherently unreliable but the conflicts of interest vary from centre to centre and cannot be pin-pointed.
I am not sure enough is made of these issues and to attempt to make statistical sense of the differences observed is not especially scientifically rigorous.
Author Response
Reviewer #1
This is an interesting retrospective posthoc registry analysis of decision making for PCI or CABG at heart team meetings in a single centre.
The authors conclude that the presence of the head of department for surgery or cardiology impacted on the decision making.
Whilst I applaud the essence of the point made here, there are very many confounding factors (some of which the authors acknowledge) with this study.
The analysis of this study was all performed retrospectively - it isnt presented clearly how many patients actually went through the process in the way stated i.e. taken off the table if anatomy was complex as described and discussed the same day; or if the discussion was deferred until the desired person was present at the heart team meeting. It isnt clear for how many patients complete data was not available or who were deemed too high risk for anything, or who needed same day treatment.
A: The answer to the first part of the question is very simple: All patients having a two-vessel disease with significant left main stenosis or a three-vessel disease underwent this procedure. The only exemption is emergency cases, and the latter information was now included in the method section. All other patients will be discussed! This discussion takes place during our daily afternoon conference, and before this conference starts it is not known whether there will be any cases to discuss nor if one of the directors will be present. We do not retain patients until the desired person attends the meeting, we had no patients with incomplete data. Also, patients being deemed too high risk for anything will be discussed in that meeting, and the decision to let them on e.g. drugs only is a joint one. And finally, also patients who state on the table that they never will undergo CABG will be taken off the table, to have a HT discussion on this case and to be able to discuss the pros and cons for the therapies available with the patients.
Despite the statement of limitations, departmental hierarchy structures are inevitably complex with multiple relationships and conflicts between individual members. It is a valid and interesting point to make that the necessity for a heart team is not evidence-based and is inherently unreliable but the conflicts of interest vary from centre to centre and cannot be pin-pointed.A: The authors fully agree to the thoughts of reviewer #1
I am not sure enough is made of these issues and to attempt to make statistical sense of the differences observed is not especially scientifically rigorous.
Reviewer 2 Report
This is an interesting study by Abdulrahman et al. examining the impact of hierarchy on multidisclipinary HT reocmmendations. The authors utilize their extensive and standardized HT approach to patient care and examine issues related to bias. I think the study is of merit and interest and highlights possible selection biases in care. The authors acknowledge all the limitations including the fact that the leadership model may be different in their institution versus those at other institutions either locally or internationally. Nonetheless, this study provides useful insights.
While the analysis and study design is valid, I have a few comments that would help strengthen the message:
Table 1, in my opinion, is essentially useless because it doesn't compare the patient profiles in the different scenarios presented in Fig 1. The 4 groups in figure 1 should be compared together and the respective patient profiles should be examined to ensure that the recommendations and risk profile matches that to the patient. It could be that the HOC and HOS are present in cases where the patients being reviewed are much more complex. It will be important to look at time trends separately for each group above to see how these have evolved. Combining all of them together is difficult to interpret. A subgoup analysis would be to stratify the cohort into risk profiles (either STS as well as SYTNAX score) and examine these biases within them. This will help us understand where the biases are influenced by the patient risk itself.Author Response
Reviewer #2
This is an interesting study by Abdulrahman et al. examining the impact of hierarchy on multidisclipinary HT reocmmendations. The authors utilize their extensive and standardized HT approach to patient care and examine issues related to bias. I think the study is of merit and interest and highlights possible selection biases in care. The authors acknowledge all the limitations including the fact that the leadership model may be different in their institution versus those at other institutions either locally or internationally. Nonetheless, this study provides useful insights.
A: The authors thank reviewer #2 for the encouraging words
While the analysis and study design is valid, I have a few comments that would help strengthen the message:
Table 1, in my opinion, is essentially useless because it doesn't compare the patient profiles in the different scenarios presented in Fig 1. The 4 groups in figure 1 should be compared together and the respective patient profiles should be examined to ensure that the recommendations and risk profile matches that to the patient. It could be that the HOC and HOS are present in cases where the patients being reviewed are much more complex. It will be important to look at time trends separately for each group above to see how these have evolved. Combining all of them together is difficult to interpret. A subgoup analysis would be to stratify the cohort into risk profiles (either STS as well as SYTNAX score) and examine these biases within them. This will help us understand where the biases are influenced by the patient risk itself.
A: The authors changed the Table 1 and present now the characteristics of the patients according to the 4 profiles as presented in Figure 1. The authors agree, that it would be helpful to have a subgoup analysis would be to stratify the cohort into risk profiles (either STS as well as SYTNAX score) and examine these biases within them. However, we also believe that this analysis can not be done with our data, mainly due to the relatively low number of patients. If we stratify according the SYNTAX score, we will end up in two groups (HOC or HOS present) with very low numbers (<10 patients per group in low and high SYNTAX score). These low numbers can not be adequately analyzed and the p-Values will be irrelevant
Reviewer 3 Report
In my opinion, this is a very interesting study that tackles an important issue. In its retrospective design, this study aimed to assess whether the hierarchy of professionals attending heart team consultation meetings affected clinical decision-making in the context of isolated multivessel disease. In their study, 209 heart team discussions were included. According to their data, when chiefs of both cardiology and cardio surgery were absent fewer CABG procedures and more PCI procedures were undertaken. Importantly, if the chief of cardio surgery was the only one present, significantly more CABG procedures were undertaken in comparison to PCI whereas when cardiology chief was only present, significantly more PCI procedures were undertaken. Finally, it seems that this trend of decision-making was preserved throughout the study period (2012-2015).
Mostly I have issues with the presentation of the results in this manuscript. This should be revised appropriately, as outlined in the comments below.
Authors have listed relevant ethical approvals and disclosures.
I would suggest authors create a new graph showing the relationship of CABG-to-PCI ratio is given respective HT scenario in the following order: none of the directors present, both directors present, cardiology chief present only, cardio surgery present only. Present Figure 1 can be retained in the manuscript.
It seems that in authors' HT meetings, there is a strong preference for cardio surgery vs. interventional cardiology across all of the domains, except for the scenario in which only interventional cardiologist is present. These are very interesting results. For example, the difference in decisions to go PCI route in case of interventional cardiologist only being present (46%) to only 17% when the only cardio surgeon is present is staggering. Could authors compare this to any literature data out there? What is usually the ratio and balance between cardiosurgical vs. interventional decisions in HT meetings? Are there any data showing similar or different results?
Figure 2. should be depicted as a 2D trend-line graph presenting CABG and PCI lines across the three studied years period and in two scenarios: only director of cardio surgery present and only director of interventional cardiology present.
Authors should create a graph showing appropriateness for CABG and PCI by using ACC AUC criteria.
It would be interesting to do a head-to-head graphical comparison on how many of CABG procedures were appropriate according to ACC AUC and SYNTAX II criteria when the only head of cardiac surgery was present and how many of PCI procedures were appropriate when the only head of interventional cardiology was present. In this way, the discrepancy between the quantified approach by using an established score vs. personal „gestalt“ could be understood and presented more clearly.
Finally, I think this is an interesting paper that has implications for clinical practice and might be better given that required changes and amendments are undertaken.
Author Response
Reviewer #3
In my opinion, this is a very interesting study that tackles an important issue. In its retrospective design, this study aimed to assess whether the hierarchy of professionals attending heart team consultation meetings affected clinical decision-making in the context of isolated multivessel disease. In their study, 209 heart team discussions were included. According to their data, when chiefs of both cardiology and cardio surgery were absent fewer CABG procedures and more PCI procedures were undertaken. Importantly, if the chief of cardio surgery was the only one present, significantly more CABG procedures were undertaken in comparison to PCI whereas when cardiology chief was only present, significantly more PCI procedures were undertaken. Finally, it seems that this trend of decision-making was preserved throughout the study period (2012-2015).
Mostly I have issues with the presentation of the results in this manuscript. This should be revised appropriately, as outlined in the comments below.
Authors have listed relevant ethical approvals and disclosures.
I would suggest authors create a new graph showing the relationship of CABG-to-PCI ratio is given respective HT scenario in the following order: none of the directors present, both directors present, cardiology chief present only, cardio surgery present only. Present Figure 1 can be retained in the manuscript.
A: The authors are grateful for this suggestion and we included a new Figure (Fig. 2) into the manuscript, providing the requested information
It seems that in authors' HT meetings, there is a strong preference for cardio surgery vs. interventional cardiology across all of the domains, except for the scenario in which only interventional cardiologist is present. These are very interesting results. For example, the difference in decisions to go PCI route in case of interventional cardiologist only being present (46%) to only 17% when the only cardio surgeon is present is staggering. Could authors compare this to any literature data out there? What is usually the ratio and balance between cardiosurgical vs. interventional decisions in HT meetings? Are there any data showing similar or different results?
A: The authors are not quite sure whether there is a surgical preference in our center. In the second paragraph of our discussion we gave some insight into international patterns of CABG utilization. For example, in three vessel disease the CABG-to-PCI-ratio is in the UK 9 but in France 0.7 (2.3 for whole Europe), and 4.9 for the US. The overall CABG-to-PCI-ratio in our center of 2.7 fits well into this range and is even more on the PCI side. It is really important to realize, that these numbers are only true for 3 vessel disease, for 1- or 2-vessel disease the numbers of patients undergoing CABG are much lower internationally and in our center. The reviewer #3 will be surprised how many patients have to be scheduled for CABG if “ACC AUC” and “SYNTAX II” is calculated for every single patient.
Figure 2. should be depicted as a 2D trend-line graph presenting CABG and PCI lines across the three studied years period and in two scenarios: only director of cardio surgery present and only director of interventional cardiology present.
A: This Figure is now labelled as Figure 3 and was changed as requested
Authors should create a graph showing appropriateness for CABG and PCI by using ACC AUC criteria.
It would be interesting to do a head-to-head graphical comparison on how many of CABG procedures were appropriate according to ACC AUC and SYNTAX II criteria when the only head of cardiac surgery was present and how many of PCI procedures were appropriate when the only head of interventional cardiology was present. In this way, the discrepancy between the quantified approach by using an established score vs. personal „gestalt“ could be understood and presented more clearly.
A: The authors included a Figure 4 which shows the appropriateness of the decisions for CAGB/PCI when only the head of surgery or only the head of cardiology was present.
Finally, I think this is an interesting paper that has implications for clinical practice and might be better given that required changes and amendments are undertaken.
Round 2
Reviewer 2 Report
Yes, the manuscript has significantly improved and now warrants publication.Author Response
Yes, the manuscript has significantly improved and now warrants publication.
Answer: The authors like to thank the reviewer for the kind suggestion
Reviewer 3 Report
I appreciate all the changes introduced by the authors based on the requests made.
However, it would additionally enrich the manuscript if the authors could comment on their results and put in perspective by including the latest results and reference from the latest SYNTAXES study (Thuijs et al. 2019 Lancet) showing the comparable 10-year survival between bypass surgery and coronary stenting. This study also showed that CABG provided a survival benefit in patients with the 3-vessel disease while this was not in the case in patients with LM lesion.
I would suggest another table to be made and provided by authors. This could be just by deleting these variables from Table 1 and including it in a separate table. This new table should be comprised of three columns: HOS present only, HOC present only, neither HOC nor HOS present.
In the rows, the following percentages should be provided
3-vessel disease, 2-vessel disease including a significant lesion of the main stem or proximal LAD, unprotected LM, SYNTAX score, EuroScore II, appropriate according to ACC AUC, appropriate according to Syntax II. It would also be interesting to add row Diabetes and eGFR if it can be calculated. For all respective columns, an inferential statistics and comparison must be made with the respective p-value provided in the 4th column.
Author Response
I appreciate all the changes introduced by the authors based on the requests made.
However, it would additionally enrich the manuscript if the authors could comment on their results and put in perspective by including the latest results and reference from the latest SYNTAXES study (Thuijs et al. 2019 Lancet) showing the comparable 10-year survival between bypass surgery and coronary stenting. This study also showed that CABG provided a survival benefit in patients with the 3-vessel disease while this was not in the case in patients with LM lesion.
Answer: We included the paper in the Discussion section and in the References
I would suggest another table to be made and provided by authors. This could be just by deleting these variables from Table 1 and including it in a separate table. This new table should be comprised of three columns: HOS present only, HOC present only, neither HOC nor HOS present.
In the rows, the following percentages should be provided
3-vessel disease, 2-vessel disease including a significant lesion of the main stem or proximal LAD, unprotected LM, SYNTAX score, EuroScore II, appropriate according to ACC AUC, appropriate according to Syntax II. It would also be interesting to add row Diabetes and eGFR if it can be calculated. For all respective columns, an inferential statistics and comparison must be made with the respective p-value provided in the 4th column.
Answer: The authors created this new table (Table 2) and performed the requested calculations. This new table is now included in the Result section.